# SAMCell: Generalized label-free biological cell segmentation with segment anything

**Alexandra Dunnum VandeLoo** [1‡*], **Nathan J. Malta**[2‡], **Saahil Sanganeriya**[2],
**Emilio Aponte**[2], **Caitlin van Zyl**[3], **Danfei Xu**[4], **Craig Forest**[2,3,5]

**1** School of Materials Science and Engineering, Georgia Institute of Technology, Atlanta, Georgia, United States of America, **2** School of Computer Science, Georgia Institute of Technology, Atlanta, Georgia, United States of America, **3** Department of Biomedical Engineering, Georgia Institute of Technology, Atlanta, Georgia, United States of America, **4** School of Interactive Computing, Georgia Institute of Technology, Atlanta, Georgia, United States of America, **5** School of Mechanical Engineering, Georgia Institute of Technology, Atlanta, Georgia, United States of America

‡ These are co-first authors of this work.
* adunnum1@gatech.edu

**Data availability statement:** All training datasets are available from the GitHub realease: https://github.com/NathanMalta/SAMCell/releases/tag/v1.

## Abstract

**Background:** When analyzing cells in culture, assessing cell morphology (shape), confluency (density), and growth patterns are necessary for understanding cell health. These parameters are generally obtained by a skilled biologist inspecting light microscope images, but this can become very laborious for high-throughput applications. One way to speed up this process is by automating cell segmentation. Cell segmentation is the task of drawing a separate boundary around each cell in a microscope image. This task is made difficult by vague cell boundaries and the transparent nature of cells. Many techniques for automatic cell segmentation exist, but these methods often require annotated datasets, model retraining, and associated technical expertise.

**Results:** We present SAMCell, a modified version of Meta's Segment Anything Model (SAM) trained on an existing large-scale dataset of microscopy images containing varying cell types and confluency. Our approach works on a wide range of microscopy images, including cell types not seen in training and on images taken by a different microscope. We also present a user-friendly UI that reduces the technical expertise needed for this automated microscopy technique.

**Conclusions:** Using SAMCell, biologists can quickly and automatically obtain cell segmentation results of higher quality than previous methods. Further, these results can be obtained through our custom Graphical User Interface, thus decreasing the human labor required in cell culturing.

## Background

Cell culture is widely used for biological research applications. In conducting cell culture, microscopy is used to assess cell attributes like confluency (density of cell growth), morphology (cell shape), and count metrics which relate to cell health and allow a researcher to assay viability visually throughout the cells' growth period [1]. These parameters, typically obtained

**Funding:** The author(s) received no specific funding for this work.

**Competing interests:** The authors have declared that no competing interests exist.

manually by a skilled biologist, are important to the success of biological experiments. Confluency, for example, is often used to gauge when cells are ready to be passaged, differentiated, or otherwise manipulated [2] and requires experience to assess accurately by eye.

Automating this assessment of cell attributes is an open area of research. Automatically segmenting cells is a difficult problem. Data annotation is especially laborious as cells routinely number in the hundreds per image. Further, due to differences in imaging parameters like contrast and field brightness, high variance in pixel intensity indicative of cell boundaries exists between images from different microscopes, even of the same imaging technique. Finally, cells are often tightly clumped together and have difficult to discern edges even for annotators. An ideal model would accurately predict cell boundaries for a wide range of images and cell types to reduce the need for manual data annotation. If accurate cell boundaries could be produced, relevant parameters to cell health, like average number of nearest cell neighbors, average cell aspect ratio, and average cell size, could be automatically calculated and reported. Reliably automating the segmentation of microscopy images across cell types and imaging parameters has not yet been achieved [3]. Driven by the needs of cell culturing, we are concerned with finding the boundaries of entire, unlabeled cells in this work. Unlike other approaches [4], we do not explore segmenting the nucleus or other regions as separate from the remainder of the cell, instead focusing on the complete cellular boundary which is most relevant for cell culture monitoring and assessment.

Interestingly, cell segmentation poses new challenges not seen in standard, multi-class segmentation like the previous Segment Anything Model (SAM) work [5]. For example, cells are often packed tightly in microscope images, with weak or even ambiguous edge features. As such, care must be taken not to merge adjacent cells in predicted masks. One approach, employed by the U-Net paper, is to segment the image into two categories, "cell" and "background", with a minimum 1-pixel border of category background between adjacent cells. Then, connected components of category "cell" are extracted from the prediction as individual cells. To avoid clumped cells being merged by the model, a pixel-wise weighted loss function with a higher weight near cell boundaries is used [6]. Unfortunately, due to the constraint that predicted cell masks cannot be in contact with each other, this approach restricts the model's ability to accurately predict the cell masks when dealing with clumped cells. Further, the light-weight U-Net model can under-fit larger datasets and has a reduced potential to benefit from pretraining compared to larger models.

Subsequent work expands on this idea, introducing a 3rd "cell border" category and experimenting with various loss function weightings. After classification, these edge pixels are assigned to the nearest cell [7]. Although this additional category eliminates the need for a background border, "cell border" category pixels can become difficult to assign to the correct cell when cells are not circular or contain fine features, like long offshoots. This approach still has limited benefit from pretraining and has the potential to under-fit large datasets. From initial experiments with Segment Anything, we observe that for cells with weak edge features, a three-category classification with a weighted loss function is insufficient to prevent the merging of clustered cells.

Another prominent cell segmentation work, Cellpose, uses a modified U-Net architecture to estimate a specialized vector field, computed from a simulated diffusion process [8]. The authors define a diffusion process that produces a vector field where, for each cell, gradient vectors point away from the cell's center in all directions. Cell masks are then computed by finding the fixed points of this vector field, locations where the vectors from neighboring cells intersect. Because Cellpose calculates and predicts this vector field independently for the x and y components, rotation in data augmentation becomes non-trivial.

In April 2023, Meta released Segment Anything Model (SAM) [5], a general model for image segmentation. SAM was trained on a diverse dataset of 11 million everyday images containing, in total, more than 1 billion segmentation masks. Due to this extensive dataset, Segment Anything is a great generalist, showing strong performance even on datasets not seen during model training. Given a point or bounding box in an image (the "prompt"), SAM can produce a reasonable segmentation boundary. Alternatively, SAM can generate masks automatically by uniformly sampling points around the image and keeping the most confident segmentations. Thanks to the extensive dataset used, SAM can produce a reasonable boundary for a wide set of objects.

Because of SAM's strong performance on natural images, there have been attempts to fine-tune it for the biomedical domain. MedSAM [9], for example, fine-tunes SAM using more than a million images collected from various medical imaging modalities, like X-Rays, Computed Tomography (CT) Scans, and ultrasounds. These authors preserve SAM's prompting capability, training MedSAM to determine masks from a bounding box. In training, the authors completely update the weights in the image encoder and mask decoder while freezing the prompt encoder. MedSAM delivers strong results, exceeding baselines in a variety of segmentation tasks and offering impressive cross-dataset generalization performance on unseen datasets.

SAMed [10], a concurrent work to MedSAM, fine-tunes SAM to segment individual organs in CT scans. Interestingly, SAMed does not require a prompt and, unlike SAM, can segment images into distinct categories, like liver, stomach, and pancreas. Further, SAMed is trained on the comparably smaller scale Synapse multi-organ CT dataset, which consists of just a few thousand images. SAMed uses Low Rank Adaptation [11] to fine-tune SAM's image encoder, while retraining all parameters in the mask decoder.

It is important to note that neither MedSAM nor SAMed were designed for cell segmentation, and therefore, directly applying these methods to cell segmentation would not be appropriate or meaningful. However, both works give important insight into successful fine-tuning strategies and architectural choices for adapting SAM in a specialized domain. Inspired by MedSAM's encouraging segmentation performance of organs, we take a similar approach of fine-tuning the SAM image encoder for our particular dataset. However, we expand upon their work by eliminating the need for a bounding box prompt, which would be impractical for cell segmentation where hundreds of cells may be present in a single image. Similarly, while SAMed demonstrates semantic segmentation without prompts, we extend this approach by conducting unprompted instance segmentation rather than being limited to discrete categories, which cannot distinguish individual cell instances when cells are touching or overlapping.

Due to the popularity of the Segment Anything Model, a few concurrent works also use it to approach the cell segmentation task. CellSAM [12], for example, first creates rectangular bounding boxes around each cell in an image with an object detection model. These bounding boxes serve as prompts for a fine-tuned SAM. Thus, CellSAM can produce cell boundaries while preserving the prompting capability of the Segment Anything Model. Another work, "Segment Anything for Microscopy" [13] introduces an extension of the default Segment Anything Model, called micro_sam. Like CellSAM, micro_sam offers a version of the Segment Anything Model fine-tuned on a dataset of microscopy tasks. Uniquely, however, micro_sam offers support for some microscopy-specific tasks: tracking cells over time in images and segmenting a cell in a 3D collection of images. It also offers integration with an existing GUI for easy manual prompting by a user. Taking a different approach, CellViT [14] leverages the pretrained image encoder from Segment Anything, while replacing SAM's mask

decoder with a U-Net-inspired segmentation decoder. CellViT is able to segment and classify cell nuclei in tissue images with impressive results.

We present SAMCell, a fine-tuned approach based on SAM for predicting cell boundaries in microscope images. Unlike the default SAM, we observe success in cases even when cells are densely packed or boundaries are soft. Like concurrent works, SAMCell inherits Segment Anything's main advantage: a Vision Transformer [15] based architecture and extensive pre-training. These advantages improve generalization and allow our method to better tackle the complex task of cell segmentation compared to other approaches.

We take a unique approach to fine-tuning compared to other concurrent works. Rather than using an object detection model for prompting (like CellSAM) or a custom decoder module (like CellViT), we disregard prompting and pose segmentation as a regression task. Our main contributions are twofold: First, we introduce a novel approach that fine-tunes SAM to output a real-valued distance map describing the Euclidean distance to a cell border for each pixel in the image. We then recover the boundary using a post-processing technique based on the watershed algorithm to effectively address the challenging problem of segmenting densely packed cells with ambiguous boundaries. Second, we create and publicly release two new annotated datasets, PBL-HEK and PBL-N2a, providing much-needed benchmark resources for evaluating cross-dataset generalization performance of cell segmentation algorithms across diverse cell morphologies and imaging conditions. These datasets contain phase-contrast microscopy images of cell lines commonly used in biological research but captured with different microscopes than those in existing training datasets, facilitating a realistic assessment of model generalization. We find that our method exceeds the performance of existing approaches on both cells similar to those seen in training (test-set) and on completely novel cell lines from images taken by other microscopes (zero-shot, cross-dataset generalization), demonstrating the robust generalization capabilities that make SAMCell particularly valuable for practical cell segmentation applications across diverse laboratory environments.

## Methods

### Datasets and evaluation approaches

We evaluate SAMCell using two distinct evaluation approaches. First, we assess standard test-set performance, where a model is trained on a portion of a dataset and evaluated on a held-out test set from the same dataset. In this scenario, the training and test images share similar characteristics since they come from the same data distribution.

Second, we evaluate generalization performance on novel datasets not seen during training. This cross-dataset evaluation, representing a zero-shot scenario, demonstrates a model's ability to generalize to new cell types and microscopy conditions. For this purpose, we created two new annotated datasets specifically designed to test model generalization across different microscopes and cell morphologies.

**Existing datasets for training and test-set evaluation.**   For model training and test-set evaluation, we utilize two established datasets of cell microscopy images. First, the LIVECell dataset [16] comprises over 5,000 phase-contrast images across 8 cell types, containing approximately 1.7 million individually annotated cells. While 5,000 images might not be considered extensive in general computer vision contexts, it represents a significant resource in the field of annotated cell microscopy, where expert annotation is particularly time-consuming and challenging. The dataset features cells with varying confluency and morphology, often with low contrast, creating a challenging segmentation task that is valuable for model evaluation.

A limitation of LIVECell is that all images were captured using the same phase-contrast microscope. This means that all images in this dataset are of a standardized size ($704 \times 520$), preventing the need to resize. However, since different microscopes produce variations in imaging parameters (contrast, brightness, resolution), this also potentially limits LIVECell's representation of the broader cell segmentation domain.

To complement LIVECell, we also employ the Cytoplasm dataset from the Cellpose authors [8]. This dataset contains approximately 600 microscopy images scraped from the internet and subsequently annotated. Though more modest in size than LIVECell, it offers valuable diversity as it encompasses various microscopy techniques, including both bright-field and fluorescent images from different microscopes. This diversity makes the Cytoplasm dataset particularly valuable for training models that need to generalize across different imaging conditions. Some of the images in the Cytoplasm dataset have 3 channels, with separate colors showing the nucleus and membrane using fluorescent labeling. Because we are specifically concerned with label-free, whole-cell segmentation in this work, we convert all images in this dataset to grayscale for training and evaluation. Because images in the Cytoplasm dataset are of different sizes, we also resize all images to $512 \times 512$, preserving the aspect ratio by adding a border as needed. A majority of images are near this size and close to square. This resizing allows us to more easily train our model and baselines by avoiding jagged arrays.

**New datasets for cross-dataset zero-shot evaluation.** To rigorously evaluate generalization capabilities, we created two new annotated datasets: PBL-HEK and PBL-N2a. Each contains 5 phase-contrast microscopy images of Human Embryonic Kidney (HEK) 293 and Neuro2a (N2a) cell lines, respectively, captured with microscopes different from those used for the training datasets. These cell lines are commonly used in biological research but were not represented in our training data. Each image contains approximately 300 cells, providing sufficient data for meaningful evaluation.

We named these datasets after our lab, the Precision Biosystems Laboratory (PBL), and the cell lines they contain. These datasets were manually annotated by an expert biologist co-author and are used exclusively for evaluation purposes. By applying models trained on the Cytoplasm dataset to these new datasets, we can assess how well the models generalize to novel microscope settings and cell morphologies—a critical capability for practical applications in diverse laboratory environments.

The differences in microscope settings and cell morphologies between our training and evaluation datasets provide a realistic test of generalization. This approach is particularly valuable since cellular imaging varies significantly across laboratories and microscope setups, making robust generalization a key requirement for broadly applicable cell segmentation tools.

## Evaluation metrics

For evaluation, we employ two cell segmentation metrics used within the Cell Tracking Challenge [17], namely, Detection Accuracy Measure (DET) and Segmentation Accuracy Measure (SEG) [18]. These metrics assess the models' accuracy for Detection (the ability to identify objects correctly) and Segmentation (the ability to match the boundaries of each object correctly), respectively, in comparison to a given "ground-truth" annotation.

The SEG metric uses a simple Jaccard index: the intersection between a ground truth annotation and a corresponding model prediction, divided by the union of these two values. The Jaccard index was calculated on matching intersections of individual segmented cells.

This is shown below, where $R$ represents a set of pixels of a "reference" (ground truth) cell, and $S$ represents a set of pixels in a "segmented" (model predicted) cell:

$$SEG = J(S, R) = \frac{R \cap S}{R \cup S}$$

The Detection Accuracy Measure (DET) [18], quantifies a model's ability to identify cells in an image without regard to the exact boundary the model produces. This is done with a graph-based approach, the "Acyclic Oriented Graph Matching Measure for Detection" (AOGM-D), which quantifies the minimum cost of transforming one graph representation into another through elementary graph operations. To compute the AOGM-D, we first generate graphs from the ground truth and model prediction, with each cell serving as a node. Then, the number of basic graph operations (e.g., add node, remove node, split node into two) required to convert the predicted graph into the ground truth graph is computed.

For normalization, the number of graph operations needed to convert an empty graph (a graph with no nodes, representing no detected cells) into the ground truth graph is also computed. This is denoted AOGM-D$_0$. Finally, a normalized metric between 0 and 1 can be computed as follows:

$$DET = 1 - \frac{\min(AOGM\text{-}D, AOGM\text{-}D_0)}{AOGM\text{-}D_0}$$

For both SEG and DET, the overall performance is defined on an interval between 0 and 1, where better performance is indicated by values closer to 1.

Following the Cell Tracking Challenge, the overall performance measure (OP$_{CSB}$) is computed by averaging the corresponding DET and SEG values:

$$OP_{CSB} = 0.5 \cdot (DET + SEG)$$

## The default segment anything model

As mentioned, the Segment Anything Model is a state-of-the-art generalist model for image segmentation. A diagram of SAM's architecture is displayed in Fig 1. SAM consists of a large image encoder, based on ViT [15], that converts a $1024 \times 1024$ image into a condensed embedding vector. Optionally, an input mask can be added to this embedding, as an additional input to the model. The image embedding is then supplied to a lightweight mask decoder along with an encoded prompt (or a default prompt embedding if no prompt is supplied). Subsequently, this mask decoder generates three sets of $256 \times 256$ binary masks, each accompanied by a "score" value denoting the model's confidence for each mask. Finally, these masks are upscaled bilinearly to match the input dimension. Multiple masks are supplied to resolve ambiguity if there are multiple reasonable segmentations for a certain prompt. Meta offers pretrained variants of Segment Anything in 3 sizes, corresponding to the size of the image encoder: Base, Large, and Huge. This creates three SAM models of varying sizes based on image encoders, referred to as SAM-base, SAM-large, and SAM-huge.

Unfortunately, when employing existing fine-tuning approaches seen in MedSAM [9] or SAMed [10], difficulties arise with issues specific to the domain of cell segmentation. Using standard 2 (cell, background) or 3 (cell, cell border, background) category segmentation with the Segment Anything architecture, we observe poor results when cells have low contrast boundaries, as is seen in Fig 2(a). We find performance to be poor even when employing pixel-wise weighted loss functions[6,7], which have been used to tackle this problem in the past. When these common characteristics arise, adjacent cells are predicted as being merged

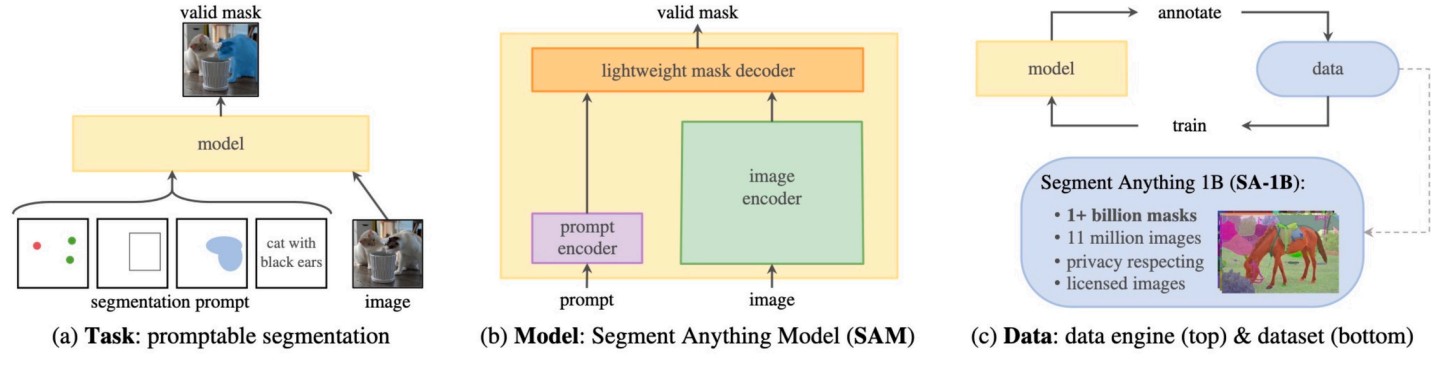

(a) **Task**: promptable segmentation     (b) **Model**: Segment Anything Model (**SAM**)     (c) **Data**: data engine (top) & dataset (bottom)

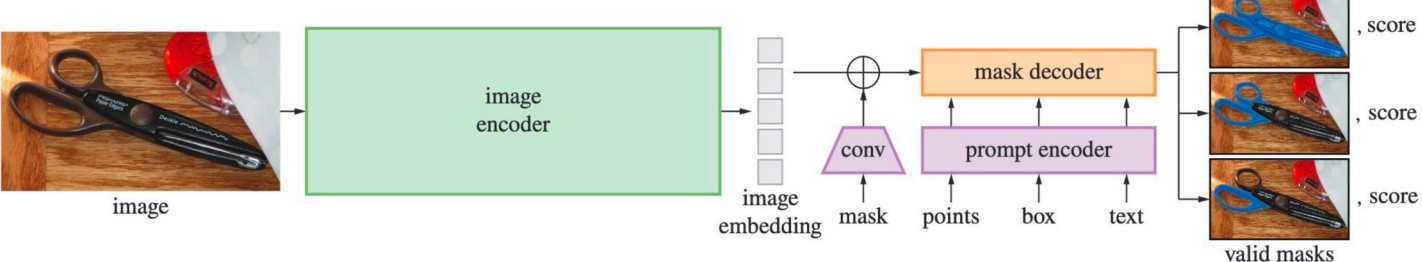

**Fig 1. Structure of the Segment Anything Model (SAM) from the original paper [5].**

together along weak edges. Additionally, these subpar segmentation results have a detrimental impact on the accuracy of insights derived from segmentation. For instance, they can lead to an underestimation of the cell count and an inflation of the average cell size when cells are merged.

## Segmentation as regression

We formulate cell segmentation as a regression problem, drawing inspiration from CellPose [8], to address difficulties with soft edges. Rather than predicting segmentation masks directly like U-Net [6], MedSAM [9], or SAMed [10], we predict a real-valued distance map, describing the Euclidean distance from each pixel to its cell's boundary (or 0 if the pixel is not part of a cell). We normalize the map such that each pixel is a real number between 0 and 1 by dividing the distance map by the maximum distance within each cell. An example of an annotation and its associated distance map is shown in Fig 2(b) and 2(c), respectively. Existing functions for efficiently producing these distance maps are present in common Python packages like SciPy [19] and OpenCV [20]. By predicting these distance maps instead of discrete categories, this problem of merged cells can be reduced. For implementation efficiency, we compute a distance map using each annotation mask in the training set before training time. Because these distance maps are invariant or almost invariant to all our data augmentation strategies (rotation, mirroring, and scaling preserve relative distances), we can avoid recomputing these maps during every training step, significantly reducing computational overhead.

This approach does have some drawbacks for cells consisting of multiple compartments. Namely, when a cell is peanut-shaped, like those formed when a cell is in the process of splitting, SamCell would consistently segment this shape as two cells, while an expert annotator may segment it either as one cell or two cells, depending on the person. While peanut-shape

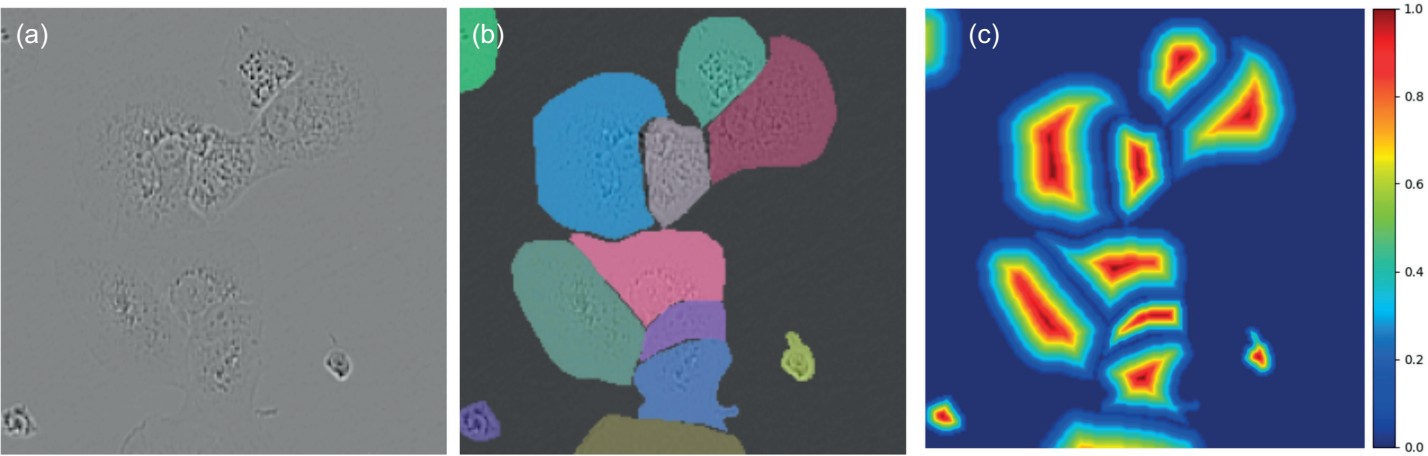

**Fig 2. (a) A cropped microscope image from the LIVECell test set, (b) overlayed ground-truth cell masks, and (c) a distance map computed from these cell masks.**

morphologies are a minority under typical cell culture conditions, we suspect that they may thus contribute to lower SEG and DET scores. For other non-circular shapes (like ovoid or "C-shaped" cells) we observe through empirical testing (data not shown) that the postprocessing technique can properly recover their geometries. It is important to note that our training and testing datasets contain realistic cell morphologies and expert annotations, without removing any problematic cases, ensuring our evaluation reflects real-world performance, so edge-cases like peanut-shaped cells are not removed from the dataset.

## Pipeline

We retain the original architecture of the Segment Anything Model (SAM) and abstract away its internal details for simplicity. More details about the model can be found in the original Segment Anything Model paper [5]. An overview of the SAMCell algorithm is provided in Fig 3. Because of VRAM constraints, concerns with inference time, and SAM's modest performance improvements with larger models, we choose to fine-tune SAM-base. To understand the effects of having a larger model capacity, we ran an ablation study comparing a model trained on SAM-Base and SAM-Large. While SAM-Large performed modestly better across all datasets and metrics, the larger model size and computational resources required make this non-ideal for user user-facing task of cell segmentation. This ablation study can be found in S1 Appendix.

**Pre-processing.** We find that applying Contrast Limited Adaptive Histogram Equalization (CLAHE) [21] is a beneficial preprocessing step. This approach eliminates any nonuniformities in brightness across the microscope field and increases the visibility of hard-to-distinguish edges. Subsequently, following SAM [5], we normalize all input images to zero-mean and unit-variance.

**Sliding window approach.** To infer SAMCell, we first separate an input image into a series of $256 \times 256$ patches using a sliding window approach, like in U-Net [6]. We use an overlap of 32 pixels on either side of these patches to avoid poor classification of areas partially in frame. To convert each $256 \times 256$ patch to the $1024 \times 1024$ input size of the image encoder, we use bilinear upsampling. Then, we pass each image through our fine-tuned

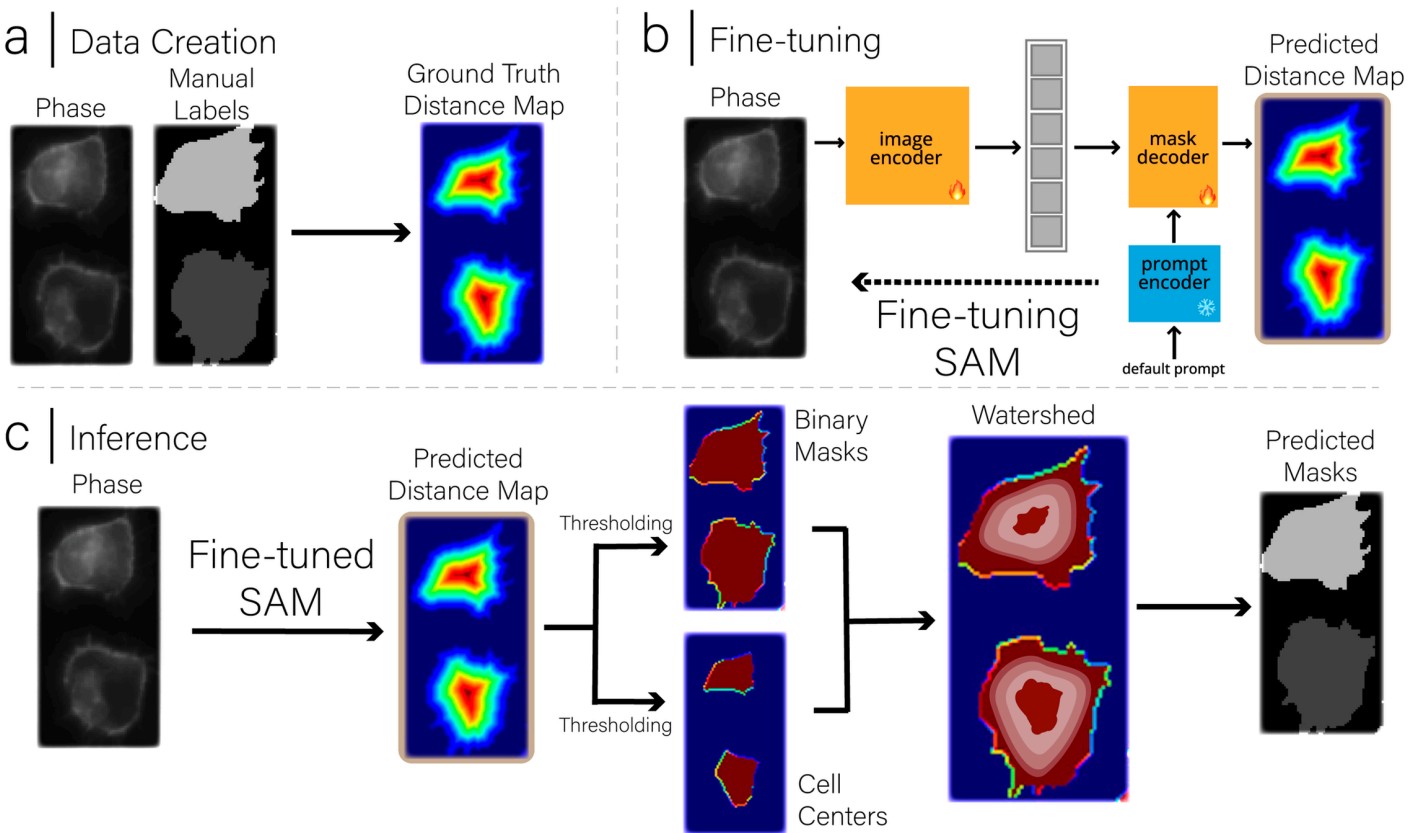

**Fig 3. Detailed Diagram of the SAMCell Pipeline, with stages (a) Ground truth distance map creation for training/fine-tuning (b) Fine-tuning of the Segment Anything Model, where we choose to fine-tune the image encoder and mask decoder and freeze the prompt decoder, and (c) our inference pipeline using the watershed algorithm.**

SAM, producing a distance map prediction. We apply the sigmoid activation function to the mask decoder output such that the output (predicted distance map) is in the correct range: [0,1].

We chose this sliding window approach over directly resizing the entire input image to $1024 \times 1024$ for two reasons. First, SAM's architecture takes $1024 \times 1024$ inputs and produces $256 \times 256$ outputs. By using $256 \times 256$ patches, we can directly compare the original patch with the model's output, without further manipulation, thus maintaining pixel-level correspondence for training and evaluation. Second, microscopy images often contain thousands of pixels in each dimension (e.g., $2048 \times 2048$ or larger). Direct resizing of such large images to SAM's required $1024 \times 1024$ input could result in a catastrophic distortion or loss of cellular detail. Our sliding window approach preserves the original resolution by consistently upsampling 256 x 256 regions to 1024 x 1024, an approach which maintains the cellular features essential for accurate segmentation.

Third, this approach aligns with SAM's original training methodology, which used $1024 \times 1024$ inputs. By maintaining the same input dimensions that SAM was pretrained on, we maximize the benefit of transfer learning from the pretrained weights. This architectural consistency is crucial for leveraging the spatial understanding SAM developed during its extensive pretraining phase.

Fourth, we hypothesized that using a patch size of $256 \times 256$ would result in optimal performance since SAM's output dimensions are $256 \times 256$. To confirm this, we ran an ablation study to confirm how patch sizes of $128 \times 128$ and $512 \times 512$ would perform. As expected, $256 \times 256$ yielded optimal performance. This study on patch sizes can be found in S1 Appendix.

Finally, the overlapping nature of our sliding window (32 pixels on each side) ensures smooth transitions between patches and prevents edge artifacts at patch boundaries, resulting in coherent segmentation across the entire image.

**Eliminating prompt.** Manually supplied prompts for densely packed cells are difficult and laborious to produce, as cells in culture often number in the hundreds per image. As such we forgo SAM's default prompting capability. To accomplish this, we employ an approach similar to that used by SAMed [10]. We freeze the prompt encoder during fine-tuning and always input SAM's default prompt as seen in Fig 3(b). Because the prompt embedding is static over fine-tuning, the mask decoder learns to predict the distance map from the image embedding exclusively.

**Finetuning methodology.** Drawing inspiration from previous SAM fine-tuning approaches [9,10], we fine-tune all parameters in SAM's lightweight mask decoder. We deviate from prior work [10] by fine-tuning all parameters in the image encoder, since we aim to infer without user-supplied prompts and therefore freeze the prompt encoder to eliminate its dependency on manual input. To address computational efficiency concerns, we explored parameter-efficient fine-tuning techniques such as Low-Rank Adaptation (LoRA) [11], which has been successfully applied in other SAM adaptations like SAMed. However, in our experiments, LoRA resulted in significantly reduced performance for cell segmentation tasks compared to full fine-tuning. Therefore, we opted for full fine-tuning of the image encoder despite the higher computational cost. This trade-off was acceptable since we used SAM-base, the smallest model variant, to minimize resource requirements while maintaining strong performance. Additionally, while training is computationally intensive, inference time remains practical for real-world use cases, especially when leveraging GPU acceleration.

**Post-processing.** Post-processing must occur to convert a predicted distance map into predicted cell masks. An outline of this process is shown in Fig 3(c). First, by thresholding the predicted distance map, we create two binary images. The first image, which we denote by "binary mask", tells us whether a certain pixel belongs to a cell or the background. To get this, we apply a threshold value denoted as *cell fill threshold* such that pixels with distance map values greater than this threshold are considered part of a cell. The second image, which we denote "cell centers", must contain exactly one connected component per cell, somewhere in the body of the cell. To get this, we apply a higher threshold value denoted as *cell peak threshold* such that only pixels with distance map values above this threshold remain, typically resulting in one connected component per cell. Although each cell type and image type would have an ideal value for these thresholds, we conducted an extensive ablation study across our two evaluation datasets to understand how these values affect performance and find an ideal set of default values. We found that while PBL-HEK and PBL-N2a have slightly different ideal threshold values, we find that a global setting of cell peak threshold = 0.47 and cell fill threshold = 0.09 yields the best average performance. More information on why these threshold values were chosen can be found in S1 Appendix.

We apply the Watershed algorithm [22], a classical approach to boundary detection, to recover cell masks from the distance map. The algorithm treats the distance map as a topographical surface where cell centers represent peaks and cell boundaries represent valleys. Starting from the cell centers identified by the cell peak threshold, the algorithm simulates

water flooding toward the simulated valleys. Cell boundaries are thus determined where the flood regions hit a clear boundary (the cell edge), or when the flood regions of different cell centers meet (if the cell edge is not clear), effectively separating individual cells even when they are densely packed. As with producing the distance map, there are existing high-speed implementations of this algorithm in common libraries like OpenCV [20] and scikit-learn [23].

The watershed algorithm was specifically chosen for our post-processing pipeline due to several key advantages in the context of cell segmentation. First, it excels at separating touching or overlapping objects, a common challenge in microscopy images where cells frequently cluster together with indistinct boundaries. The algorithm's flooding-based approach naturally identifies the boundaries between adjacent cells at points where the distance values from neighboring cell centers meet.

Second, the watershed algorithm is particularly well-suited for processing distance maps because it inherently operates on topographical representations. By treating our normalized distance map as a topographical surface where cell centers form peaks and boundaries form valleys, the watershed transformation creates a natural segmentation along these valleys. This conceptual alignment between distance maps and watershed segmentation creates a synergistic post-processing pipeline.

Third, the algorithm is deterministic and requires minimal parameter tuning once the initial distance map thresholds are established. This characteristic makes it reliable across various cell types and imaging conditions, providing consistent results when applied to similar images. Additionally, the watershed algorithm is computationally efficient compared to more complex segmentation approaches, making it suitable for processing large microscopy datasets where hundreds of cells may be present in a single image.

While we found the watershed algorithm to be optimal for our application, we also evaluated several alternative post-processing approaches:

- **Connected Component Analysis:** A simpler approach that directly labels connected regions in a thresholded binary mask. While computationally efficient, this method struggles to separate touching cells, leading to under-segmentation in dense regions—a critical limitation for accurate cell counting and morphology analysis.
- **Marker-Controlled Segmentation:** A refinement of the watershed approach that uses predefined markers to guide the segmentation process. This method can potentially reduce over-segmentation but requires more precise marker generation, which proved challenging to generalize across diverse cell types.
- **Level Set Methods:** These iterative curve evolution techniques can precisely delineate cell boundaries but are computationally expensive and sensitive to initialization parameters.
- **Graph Cuts:** This optimization-based segmentation approach treats pixels as nodes in a graph and finds optimal cuts to separate foreground from background. While powerful for some segmentation tasks, it requires significant parameter tuning for different cell types and is less effective at separating clustered cells than the watershed algorithm.

We determined that the watershed algorithm provided the best balance of accuracy, computational efficiency, and generalizability across different cell types and imaging conditions. Its ability to naturally handle the topographical nature of distance maps and effectively separate touching cells made it the optimal choice for our distance map-based cell segmentation approach.

### Data augmentation

To encourage generalization, we employ data augmentation during training as follows:

- Random horizontal mirroring
- Random rotation between –180° and 180°
- Random rescaling between 80% and 120% of the original image dimensions (preserving aspect ratio)
- Random brightness adjustment between 95% and 105% of original
- Randomly invert the image

Although not often seen in prior works, we find random inversion of the image to help encourage generalization. Certain microscopy approaches (e.g., dark field) show bright cells on a darker background. Other approaches (e.g., phase contrast) yield images with cells as a darker color on a brighter background. We use random inversion to create invariance towards this difference, to create a model that can interpret images from a range of microscopy methods. Once data augmentation has been applied to an image, a random $256 \times 256$ patch of the image and associated distance map (augmented by mirroring, rotation, and rescaling only) is used for fine-tuning. As such, every epoch, SAMCell is trained on one unique, augmented 256x256 patch from every image, ensuring maximum data utilization while maintaining training variety through the randomized patch selection and augmentation process.

### Training protocol

For all our models, we trained using early stopping over a range of 35 to 100 epochs, with a patience of 7 epochs and a minimum improvement threshold of 0.0001. We save the model after every checkpoint and take the version with the lowest loss. For SAMCell-Generalist (SAM-B), the highest performing SAM-B variant, training terminated at 35 epochs after 5 hours due to this criterion. We used a batch size of 8 on an NVIDIA A100 GPU with 80 GB of VRAM. Starting from the pretrained SAM-base model, we employed the AdamW [24] optimizer with an initial learning rate of 0.0001 and a weight decay of 0.1, setting $\beta_1 = 0.9$ and $\beta_2 = 0.999$. Following the protocol in SAMed [10], we applied a learning rate warm-up [25,26] with a period of 250 iterations, followed by a linear decay to zero over the remaining training period. For SAMCell-Generalist, we trained SAM-base as a generalist model by concatenating the LIVECell and Cytoplasm datasets. Additional experiments exploring various dataset combinations, SAM variants, and initialization strategies are discussed in the ablation studies conducted in S1 Appendix. Since the task is framed as regression to a distance map, we optimized the model using L2 loss.

## Results and discussion

### Dataset visual comparison

Fig 4 illustrates key visual differences between our training datasets (Cytoplasm and LIVECell) and our evaluation datasets (PBL-HEK and PBL-N2a). These visual differences highlight why zero-shot, cross-dataset evaluation provides a meaningful test of model generalization capabilities. The PBL-HEK and PBL-N2a datasets differ substantially in cell morphology, microscope contrast, and field brightness compared to the training datasets. PBL-HEK features densely packed cells with irregular morphologies, while PBL-N2a contains more circular cells with distinct boundaries. These differences in cellular appearance and imaging conditions make these datasets particularly valuable for assessing how well models trained on one set of microscopy images can generalize to new cellular environments.

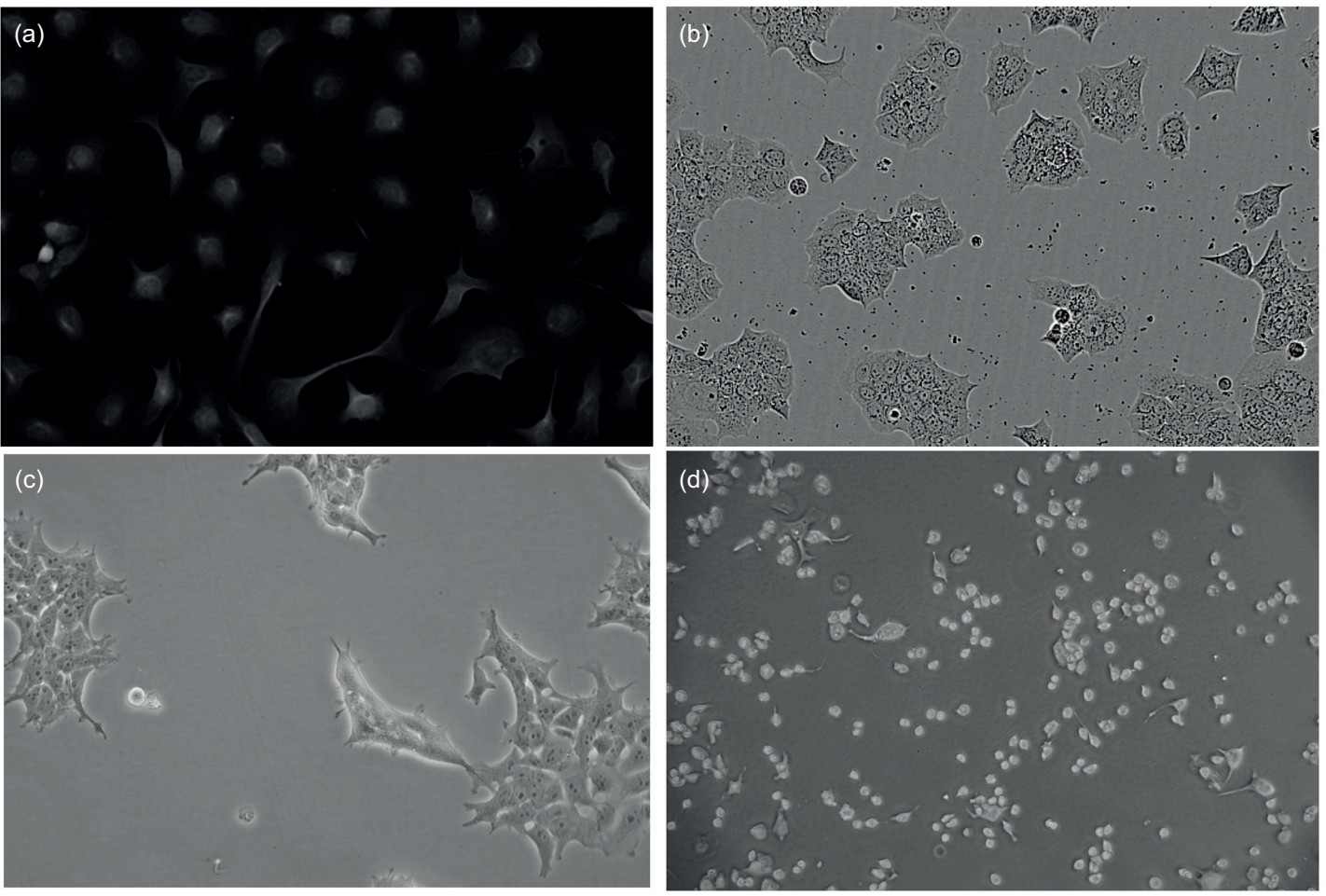

**Fig 4. Example images from the (a) Cytoplasm dataset and (b) LIVECell dataset used for training and example images from each of our evaluation datasets, (c) PBL-HEK and (d) PBL-N2a.**

## Comparison to default SAM

Although the off-the-shelf Segment Anything Model demonstrates impressive performance on a variety of tasks, we observe some common failure modes that can limit its usefulness to microscopy. We illustrate these downfalls in Fig 5. When attempting to segment densely packed cells in "automatic mask generation" mode (sampling prompt points in a uniform grid), SAM can often segment large clumps, rather than individual cells. Further, we find that SAM is prone to segmenting empty portions of the cover slip. Although this shortcoming can be overcome by a biologist prompting SAM for every cell, this introduces a manual, tedious process.

These failures likely stem from SAM's training set, which, although large in scale, is focused on everyday images like segmenting food in a supermarket or people in a sporting event [5]. Cell segmentation is a unique task: cells routinely have irregular shapes and are often clumped together with difficult-to-discriminate boundaries. As such, SAM does not have a strong prior for the appearance of cells because of its more generic dataset. This leads to SAM predicting boundaries in images that often do not correspond to individual cells.

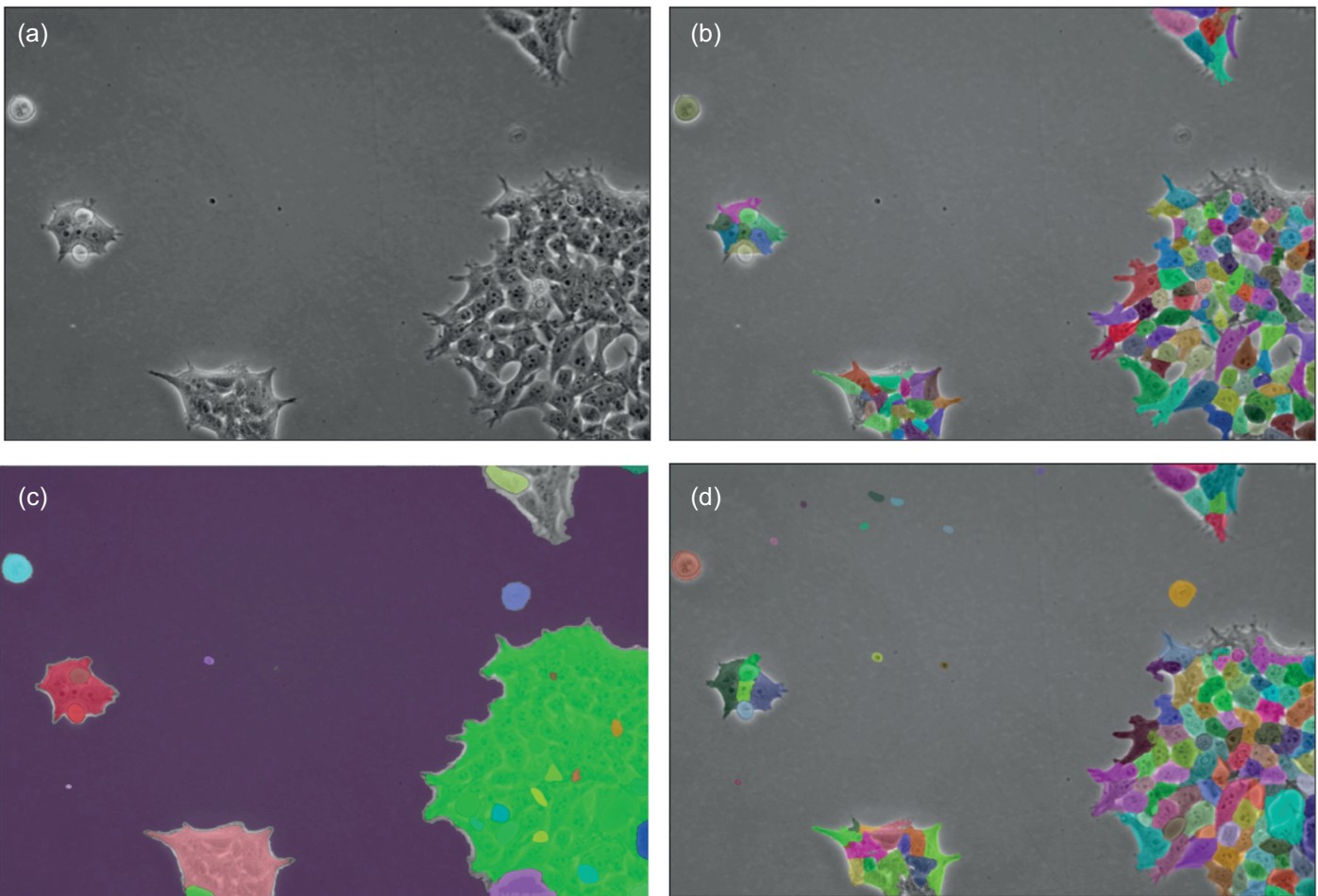

**Fig 5. (a) A single Phase Contrast image of HEK cells from our PBL-HEK dataset, (b) annotated manually by an expert, (c) automatically annotated by Meta's SAM-huge model, and (d) automatically annotated by our SAMCell model.**

Our fine-tuning of Segment Anything effectively resolves this shortcoming. During fine-tuning, the model was trained on a large corpus of cell masks in microscopy images. This improves its performance on related tasks, like the image in Fig 5(d).

## Baseline methods

In addition to the default Segment Anything Model, we select 3 existing cell segmentation models as baselines for comparison:

*Stardist* [27]: Stardist predicts cell shape as a series of line segments radiating from the cell's center point. These line segments are evenly spaced around the center and variable in length. The ends of these line segments are connected to form the boundary of a cell. This approach leverages a modified U-Net architecture to predict the component line segments for segmentation. We train this model for 50 epochs on each dataset, with the default 32 line segments per cell.

*Cellpose* [8]: Cellpose attempts to predict specialized x and y gradients from an input image. Then, these gradients are combined to produce a smooth increase in value from the

cell center to the edges. The watershed algorithm is used to recover cell boundaries from these gradients. Like Stardist, a modified U-Net architecture is used as a backbone for predictions of, in this case, image gradients. Pretrained models from the Cellpose authors exist for both the LIVECell and Cytoplasm datasets. As such, we simply use these weights without modification. These pretrained models are available at https://cellpose.readthedocs.io/en/latest/models.html.

*CALT-US* [28]: To provide an additional point for comparison, we select a high-performing model from the Cell Tracking Challenge [17]. This model uses a novel loss function, adding a regularization term to the cross-entropy loss to improve model predictions around the boundaries of cells. Using this loss function, the authors train a U-Net [6] to classify each pixel in an image into one of four categories: background, cell, or cells touching and gap between cells. Finally, the authors post-process the U-Net output to obtain cell boundaries. Conveniently, the authors provide an open-source implementation in a software package called jcell (https://jcell.org/). For the baseline used in this paper, we train for 100 epochs with a batch size of 32 and all the default parameters in jcell.

## Test-set performance

As shown in Table 1, SAMCell demonstrates strong test-set performance, surpassing baseline performance on both the LIVECell and Cellpose Cytoplasm test sets. We hypothesize this success can be attributed to two key factors: Firstly, SAMCell inherits SAM's image encoder which is pretrained on a dataset of 11 million diverse, everyday-life images. This pretraining provides SAMCell with a strong prior for objects in general, helping the model to better detect features that contribute to a cell boundary while ignoring extraneous features like field brightness or microscope contrast.

Secondly, SAMCell's Vision Transformer (ViT) architecture [15] offers distinct advantages over Convolutional Neural Network (CNN) based architectures used in our baseline methods. The self-attention mechanism in ViT enables the model to capture long-range dependencies across the entire image, which is particularly valuable for identifying cell boundaries in complex microscopy images. This architectural advantage allows SAMCell to better recognize contextual patterns that define cell boundaries, even when those boundaries have low contrast or when cells are densely packed. While CNN-based approaches can certainly be scaled up to increase capacity, the fundamental attention mechanism in transformer architectures provides a more effective way to model the relationships between distant pixels that form coherent cell boundaries.

**Table 1. A performance comparison between SAMCell and 3 existing cell segmentation baselines.** The Stardist, Cellpose, and CALT-US models were trained only on LIVECell-train then tested on LIVECell-test (top) and separately trained only on Cyto-train, then tested on Cyto-test (bottom). SAMCell, inheriting SAM's pretraining, was fine-tuned only on LIVECell-train then tested on LIVECell-test (top) and separately fine-tuned only on Cyto-train and then tested on Cyto-test (bottom).

| | SAMCell | Stardist | Cellpose | CALT-US |
|---|---|---|---|---|
| Train & Test Dataset: *Livecell* | | | | |
| SEG | **0.651876** | 0.572233 | 0.588850 | 0.559680 |
| DET | **0.892543** | 0.770644 | 0.778827 | 0.789718 |
| OP$_{CSB}$ | **0.772210** | 0.671439 | 0.683839 | 0.674699 |
| Train & Test Dataset: *Cellpose Cytoplasm* | | | | |
| SEG | **0.611144** | 0.557178 | 0.579542 | 0.415172 |
| DET | **0.865908** | 0.774016 | 0.748702 | 0.571518 |
| OP$_{CSB}$ | **0.738526** | 0.665597 | 0.664122 | 0.493345 |

## Cross-dataset generalization and zero-shot performance

As with test-set evaluations, we notice impressive cross-dataset generalization capabilities with SAMCell seen in Table 2. This cross-dataset evaluation represents a zero-shot scenario for the model, as it must segment cell types (HEK and N2a) and microscope conditions it has never encountered during training. We observe our method outperforming baseline approaches for both our PBL-HEK and PBL-N2a datasets. We speculate that this advantage comes from a combination of our distance map with the watershed approach, SAM's ViT architecture, and SAM's significant pretraining corpus. This pretraining provides SAMCell with prior knowledge of boundaries in non-microscopy images, which may help determine cell boundaries. The advantage of pretraining is further supported by our ablation study in S1 Appendix, where we show that there is a significant performance difference between using random initialization compared to the pretrained weights, even after model convergence. We also conclude that SAMCell can generalize best when trained on multiple concatenated datasets rather than a single dataset, which is supported by our ablation study in S1 Appendix.

Across all approaches, we observe much higher values for PBL-N2a compared to PBL-HEK. We attribute this to differences in appearance between the two cell lines. Neuro-2a (N2a) cells contain a circular morphology with high contrast borders, as shown in Fig 4(d). Further, these cells are less likely to grow in tightly packed clumps. Conversely, Human Embryonic Kidney (HEK) 293 cells are prone to grow in more densely packed regions and are less circular, as seen in Fig 4(c). These differences result in PBL-HEK producing a more difficult segmentation task, and thus lower SEG and DET metrics.

The qualitative comparisons in Fig 6 provide visual evidence of SAMCell's superior zero-shot, cross-dataset performance. For the challenging PBL-HEK sample, SAMCell accurately delineates individual cells in densely packed regions where Cellpose tends to merge adjacent cells or miss cells entirely. For the PBL-N2a sample, SAMCell's boundaries more closely align with the ground truth, especially for irregular morphologies. SAMCell better preserves the irregular morphology of these cells, while Cellpose's segmentations exhibit over-regularization, forcing cells into more circular shapes than their actual form. These visual results support our quantitative findings and demonstrate SAMCell's ability to handle diverse cell morphologies and densities without specific training on these cell types. The improved segmentation quality can be attributed to SAM's extensive pretraining combined with our distance map regression approach, which together enable more precise boundary detection even in challenging zero-shot, cross-dataset scenarios.

Table 2. **A zero-shot, cross-dataset performance comparison between SAMCell-Generalist (fine-tuned on Cellpose Cytoplasm and LIVECell datasets), SAMCell-Cyto (fine-tuned on Cellpose Cytoplasm dataset), SAMCell-LIVECell (fine-tuned on LIVECell dataset), and baselines trained on the Cellpose Cytoplasm dataset.**

| | SAMCell Generalist | SAMCell Cyto | SAMCell LIVECell | Stardist Cyto | Cellpose Cyto | CALT-US Cyto |
|---|---|---|---|---|---|---|
| Test Dataset: *PBL-HEK* | | | | | | |
| SEG | **0.425140** | 0.295300 | 0.298700 | 0.141827 | 0.252739 | 0.070721 |
| DET | **0.771509** | 0.613500 | 0.634500 | 0.235729 | 0.387731 | 0.161191 |
| OP$_{CSB}$ | **0.598325** | 0.454400 | 0.466600 | 0.188778 | 0.320235 | 0.115956 |
| Test Dataset: *PBL-N2a* | | | | | | |
| SEG | **0.706734** | 0.696400 | 0.569500 | 0.596675 | 0.642347 | 0.458429 |
| DET | **0.941132** | 0.918400 | 0.871100 | 0.851225 | 0.885298 | 0.632555 |
| OP$_{CSB}$ | **0.823933** | 0.807400 | 0.720300 | 0.723950 | 0.763823 | 0.545490 |

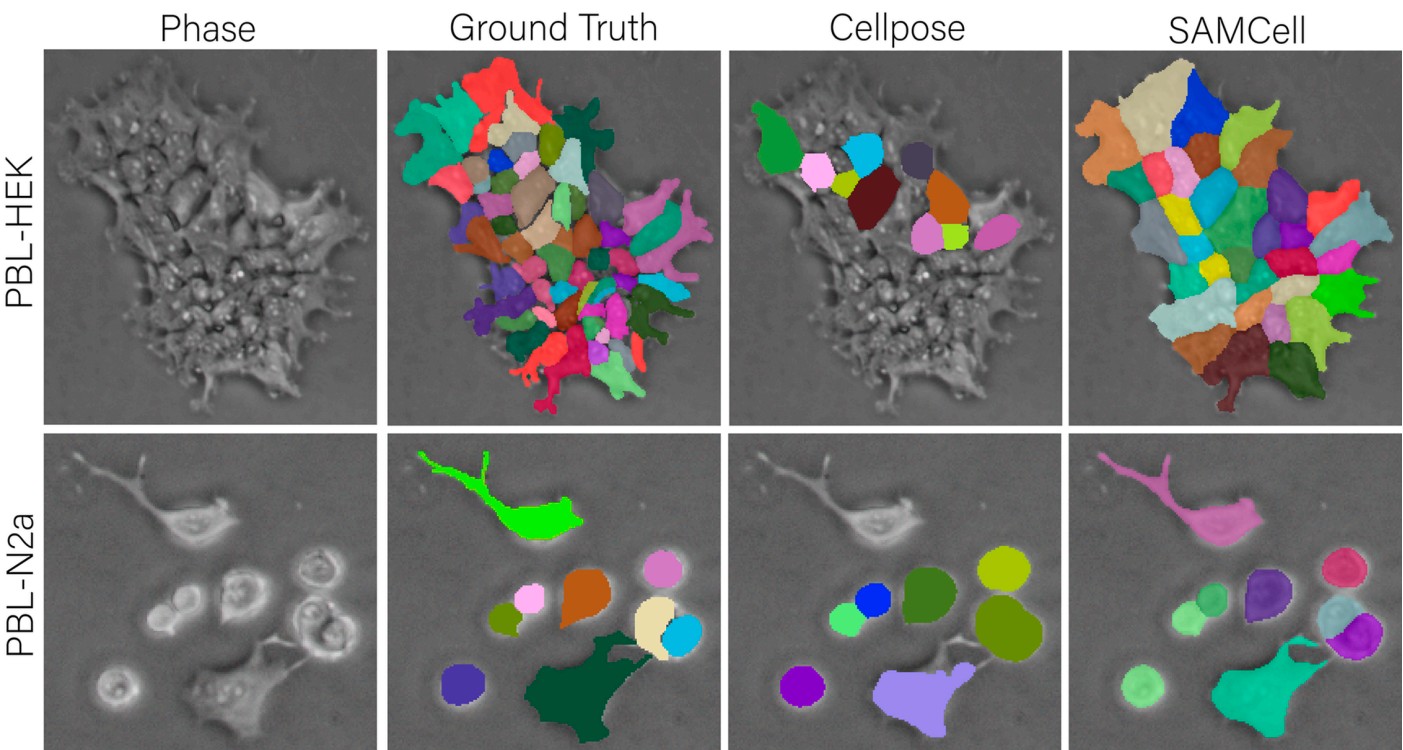

**Fig 6. Qualitative comparison of zero-shot, cross-dataset performance.** The top row shows segmentation of PBL-HEK cells (densely packed, irregular morphology), while the bottom row shows PBL-N2a cells (more circular with distinct boundaries). For each dataset, we compare ground truth annotation (left) against predictions from Cellpose-Cyto (middle), our best-performing baseline, and SAMCell-Generalist (right).

## Limitations

The main strength of the Segment Anything Model architecture is its large, ViT-based image encoder, but this increases computational requirements compared to other lighter-weight baselines. Because of the model's size, it can capture complex features and achieve high performance on large and diverse datasets - in both the original training set [5], and the microscopy datasets used in this work. Unfortunately, running inference on a larger model is more computationally expensive. With a high-end consumer GPU, the Nvidia RTX 4090, we find that running SAMCell on a single image takes about 5 seconds. We find that under the same conditions, our baselines run inference in well under a second. This is likely because our baselines are backed by the much slimmer U-Net architecture which requires reduced computational resources to run.

We find this difference is exacerbated when a GPU is not available, as would be the case on lower-end computers. We find that running SAMCell on CPU (an AMD Ryzen 7 7700X) takes approximately 2 minutes and 20 seconds per image. We observe that our lighter-weight baselines can run inference in just a few seconds under these conditions. As such, we recommend running SAMCell on a higher-end computer with a GPU to mitigate this limitation.

## User interface

**Graphical user interface.** To make SAMCell more accessible to users without a machine learning background, we create a user-friendly front-end for SAMCell. Using this, users

can click and drag microscopy images into a straightforward window and see results from SAMCell's automatic segmentation. The workflow of the user interface is shown in Fig 7. The landing page invites a user to drag and drop microscopy images as shown in Fig 7(a). Then, as shown in Fig 7(b), users can select and view images to process with SAMCell. Images in the list that have been processed are denoted green, and images being currently processed are highlighted in yellow. After an image is processed, segmentation masks can be visualized as seen in Fig 7(c). Finally, we supply a few relevant metrics extracted from the segmentation result to aid biologists in culturing cells: number of cells detected, average cell area, confluency, and number of neighbors for each cell. A table with this information is presented to the user, as seen in Fig 7(d). As mentioned in Sect , we recommend a computer with a GPU to use SAMCell, to ensure images are processed quickly. If a local GPU is not available,

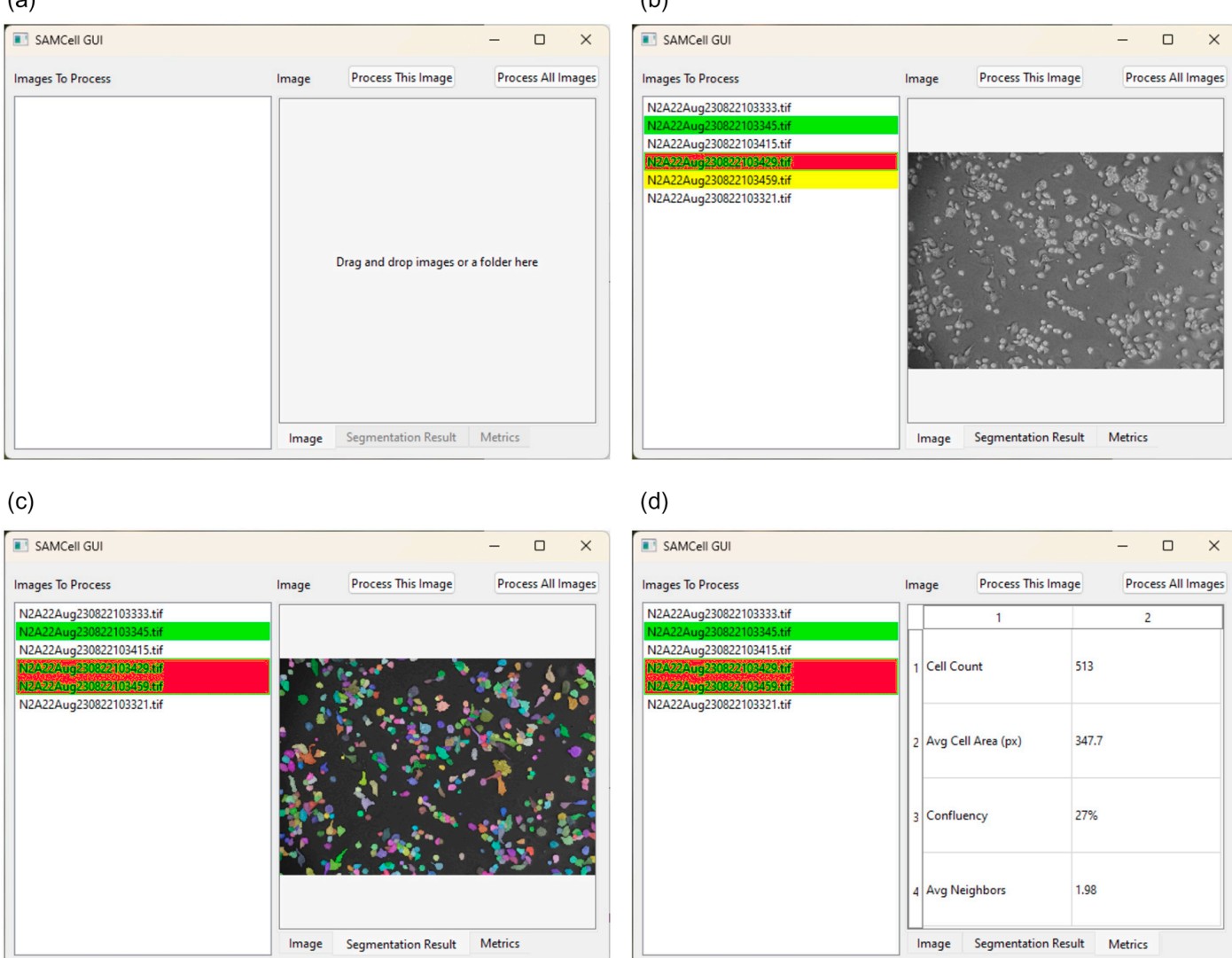

**Fig 7. A demonstration of SAMCell's interface throughout the various stages of use: (a) landing page for image upload, (b) image selection and processing status, (c) visualization of segmentation masks, and (d) extracted metrics display.**

we recommend using a cloud GPU via our Google Colab notebook (https://colab.research.google.com/drive/1016jr1JTtSI4kUIaHmnmXn290n-SM2eP?usp=sharing), though this takes slightly more expertise than a simple Graphical User Interface.

## Optimal configuration

To determine the optimal configuration for SAMCell, we conducted several ablation studies analyzing the effect of different model parameters on performance. Our experiments identified the following optimal settings:

**Model Variant:** SAM-Large consistently outperforms SAM-Base across all metrics on zero-shot, cross-dataset evaluation. However, since SAM-Base offers a good compromise between performance and computational efficiency, we elect to use SAM-Base throughout our evaluations. For applications requiring optimal accuracy, SAM-Large is available.

**Patch Size:** A patch size of 256×256 yields the best performance, particularly on challenging datasets like PBL-HEK. This aligns with SAM's native output dimensions, minimizing interpolation errors during processing.

**Pretraining:** Our experiments confirm that pretraining is crucial for performance. Models initialized with SAM's pretrained weights significantly outperform randomly initialized models, highlighting the transferable knowledge gained from SAM's diverse pretraining dataset.

**Post-Processing Thresholds:** Optimal threshold values of 0.47 for cell peak and 0.09 for cell fill were identified, achieving the best overall performance across datasets.

**Training Dataset:** Training on a combination of datasets (SAMCell-Generalist approach) yields superior zero-shot, cross-dataset performance compared to training on any single dataset.

Detailed analysis and extended results from these ablation studies can be found in S1 Appendix.

## Conclusion

We present SAMCell, a cell segmentation model based on the Segment Anything Model architecture. Using our model, we observe state-of-the-art performance exceeding powerful and commonly cited baselines like Cellpose and Stardist. We evaluate our model for two main approaches: standard test-set performance on held-out data from the same distribution as training, and zero-shot, cross-dataset performance to novel cell types and microscopes not seen during training. In both scenarios, our approach exceeds existing models in quantitative evaluation. To make SAMCell more accessible, we present a user-friendly interface to make this powerful model available to biological researchers without a machine learning background.

For future work, we identify several promising research directions: (1) Further exploration of our post-processing methodology to improve segmentation of cells with complex morphologies like dividing cells; (2) Applying our distance map regression approach to lighter-weight models like U-Net to create more computationally efficient alternatives; (3) Investigating the use of SAMCell for time-series cell tracking in live imaging; (4) Extending the model to handle 3D cell segmentation in volumetric microscopy data; and (5) Exploring additional fine-tuning approaches that might further enhance cross-dataset generalization capabilities for even more diverse microscopy techniques.

We supply training and evaluation code at https://github.com/saahilsanganeriya/SAMCell for SAMCell, as well as our user interface at https://github.com/NathanMalta/SAMCell-GUI

on Github. Additionally, we provide the trained weights for SAMCell, as well as our custom datasets (PBL-HEK and PBL-N2a) at https://github.com/saahilsanganeriya/SAMCell/releases/tag/v1.

## Supporting information

**S1 Appendix. Ablation studies.** Extended experimental details and results for training dataset configurations, model variant comparison (SAM-Base vs SAM-Large), patch size analysis, pretraining vs random initialization, post-processing thresholds, and training progression.
(PDF)

**S1 File. Appendix figures.**
(ZIP)

## Author contributions

**Conceptualization:** Alexandra Dunnum VandeLoo.

**Data curation:** Alexandra Dunnum VandeLoo, Nathan J. Malta, Saahil Sanganeriya, Emilio Aponte, Caitlin van Zyl.

**Formal analysis:** Alexandra Dunnum VandeLoo, Nathan J. Malta, Emilio Aponte.

**Investigation:** Alexandra Dunnum VandeLoo.

**Methodology:** Alexandra Dunnum VandeLoo, Nathan J. Malta.

**Project administration:** Alexandra Dunnum VandeLoo.

**Software:** Nathan J. Malta.

**Supervision:** Alexandra Dunnum VandeLoo, Danfei Xu, Craig Forest.

**Validation:** Alexandra Dunnum VandeLoo, Nathan J. Malta, Saahil Sanganeriya.

**Visualization:** Alexandra Dunnum VandeLoo, Nathan J. Malta, Saahil Sanganeriya, Emilio Aponte.

**Writing – original draft:** Alexandra Dunnum VandeLoo, Nathan J. Malta.

**Writing – review & editing:** Alexandra Dunnum VandeLoo, Nathan J. Malta, Saahil Sanganeriya, Emilio Aponte, Craig Forest.

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
