## [Decision Letter · Decision Letter 0]

31 Mar 2025

PONE-D-25-05890SAMCell: Generalized Label-Free Biological Cell Segmentation with Segment AnythingPLOS ONE

Dear Dr. VandeLoo,

Thank you for submitting your manuscript to PLOS ONE. After careful consideration, we feel that it has merit but does not fully meet PLOS ONE’s publication criteria as it currently stands. Therefore, we invite you to submit a revised version of the manuscript that addresses the points raised during the review process.

We look forward to receiving your revised manuscript.

Kind regards,

Krishnendu Sinha, Ph.D.

Academic Editor

PLOS ONE

2. Please update your submission to use the PLOS LaTeX template. The template and more information on our requirements for LaTeX submissions can be found at http://journals.plos.org/plosone/s/latex"

Additional Editor Comments (if provided):

Reviewers' comments:

Reviewer's Responses to Questions

**Comments to the Author**

1. Is the manuscript technically sound, and do the data support the conclusions?

Reviewer #1: Yes

Reviewer #2: Partly

2. Has the statistical analysis been performed appropriately and rigorously? 

Reviewer #1: Yes

Reviewer #2: N/A

3. Have the authors made all data underlying the findings in their manuscript fully available?

Reviewer #1: Yes

Reviewer #2: Yes

4. Is the manuscript presented in an intelligible fashion and written in standard English?

Reviewer #1: Yes

Reviewer #2: Yes

5. Review Comments to the Author

Reviewer #1: The paper introduces SAMCell, a modified version of Meta’s Segment Anything Model (SAM), designed for label-free biological cell segmentation. The authors aim to address the challenges of cell segmentation in microscopy images, such as vague cell boundaries and the transparent nature of cells. SAMCell is trained on large-scale datasets of microscopy images and demonstrates strong performance across various cell types and microscopy techniques. Additionally, the authors provide a user-friendly interface to make SAMCell accessible to biologists without machine learning expertise.

The experimental results demonstrate the algorithm's superiority. However, there are some weaknesses in this paper.

1. To enhance the paper, the authors should address the computational cost issue by exploring optimization techniques or lighter model variants that maintain performance while reducing resource requirements.

2. Expanding the ablation study to include more comprehensive comparisons with random initialization would better highlight the value of pretraining.

3. The authors should broaden the evaluation to include a wider range of microscopy techniques to demonstrate SAMCell’s versatility.

4. A detailed discussion of the post-processing steps, including the rationale behind the watershed algorithm choice and potential alternatives, would strengthen the methodology section.

Reviewer #2: The paper modifies Segment Anything Model (SAM) for cell segmentation task of phase-contrast and brightfield microscopic images. In addition they publish their own dataset two cell lines. I like the idea of using distance maps as prediction task, and evaluation on the own datasets.

Here are the comments:

1. I do not agree with the claim saying, "...being based on U-Net, the model may under-fit larger datasets and has a reduced advantage from pretraining.". U-Net is an architecture where any large backbone network can be used. If underfitting is the problem, the straightforward solution would be to simply increase the network size, use a larger CNN, add more layers, etc.

2. In my opinion, the number of parameters is never one of the main advantages. "...SAMCell inherits Segment Anything’s main advantages: a high parameter, ..."

3. In the Introduction section, contributions of the paper should be highlighted.

1. Prediction of distance-transform map

2. Creation and publication of novel dataset.

4. LIVECell (5000 images) and Cytoplasm (600 images) are not that large datasets.

5. Figures in general lack details. The network details should be also visible from the architecture figure.

6. In the Sliding Window Approach, what is the reasons of cropping out 256x256px patches from the input images, then upsample to 1024x1024px? Is it better than directly upsampling the input images to 1024x1024px ?

7. What is the reasons to stop on 40th epoch? What if you train more?

8. The paper need more ablation studies:

1. Different sized patches.

2. Different threshold values in the post-processing.

3. Segmenting examples of different shaped cells (dividing, c-shape etc)

4. How the segmentations changed with each (or every 10) epochs?

9. Table 1 seems unfair since SAMCell should be finetuned, and the baselines are trained from the scratch (or were they finetuned also?).

10. For Table 2, it is still better to show the scores of the baselines trained on the LIVECELL dataset. I believe they all have pretrained weights, also.

11. It is more comprehensive if segmentation comparisons from baselines and two-SAMCell models are shown.

12. User interface seems nice. It is also better to consider creating a plugin for image viewers, such as Napari or Imagej.

"Overall, the paper is well-written. However, there are still some gaps that need to be addressed to reproduce the results, especially inside the figures. In my opinion, there are not enough ablation studies to support the hyperparameter settings and other decisions, and the comparisons with the baselines are insufficient to support the claims. In its current version, this paper is more suitable to be submitted to a conference (could be top-tier).

6. PLOS authors have the option to publish the peer review history of their article (what does this mean?). If published, this will include your full peer review and any attached files.

Reviewer #1: No

Reviewer #2: No

---

## [Author Response · Author response to Decision Letter 1]

23 Apr 2025

Please refer to attached document "Response to reviewers" for better formattiing.

We thank the reviewers for their constructive feedback and valuable suggestions. Their comments have helped us improve the manuscript by making significant changes throughout and clarifying several important aspects of our work. In the marked up manuscript, additions to the text are colored in blue, and deletions are colored in red. Below, we address each point raised by the reviewers.

Reviewer #1: The paper introduces SAMCell, a modified version of Meta’s Segment Anything Model (SAM), designed for label-free biological cell segmentation. The authors aim to address the challenges of cell segmentation in microscopy images, such as vague cell boundaries and the transparent nature of cells. SAMCell is trained on large-scale datasets of microscopy images and demonstrates strong performance across various cell types and microscopy techniques. Additionally, the authors provide a user-friendly interface to make SAMCell accessible to biologists without machine learning expertise.

The experimental results demonstrate the algorithm's superiority. However, there are some weaknesses in this paper.

1. To enhance the paper, the authors should address the computational cost issue by exploring optimization techniques or lighter model variants that maintain performance while reducing resource requirements.

We appreciate this suggestion and have clarified our approach to managing computational costs in the manuscript. We have added this information to Section 2.5.4 Finetuning Methodology “To address computational efficiency…” In summary, after running an ablation study between SAM-Large and SAM-Base (Appendix A.2), we selected SAM-base, the smallest model variant, to minimize computational requirements while maintaining performance. We explored parameter-efficient fine-tuning approaches like LoRA but found they significantly reduced performance for cell segmentation tasks. Our current implementation delivers acceptable inference times, especially when using CUDA-enabled GPUs or cloud computing resources like Google Colab. 2. Expanding the ablation study to include more comprehensive comparisons with random initialization would better highlight the value of pretraining.

Thank you for this suggestion. Upon further reflection, we agreed with the need for more precise ablation studies and actually upon performing them, we were able to optimize our parameters even further. This resulted in higher perfomance than previously seen, so we would like to humbly thank our reviewer for driving this change. We have included a comprehensive comparison between SAMCell with pretrained weights and with random initialization (SAMCell-rand) in our deeper ablation study in appendix A.4 which demonstrates the substantial performance gains from pretraining. We demonstrate performance on test and zero shot datasets, as well as the effect of pretraining on the ability of the model to converge. This comparison highlights the critical role that pretraining plays in our model's effectiveness. We have also conducted four other ablation studies, all in appendix A.

3. The authors should broaden the evaluation to include a wider range of microscopy techniques to demonstrate SAMCell’s versatility.

Thank you for the thoughtful suggestion. We would absolutely consider expanding our evaluation in future work. Unfortunately, due to the scarcity of properly annotated data sets, and the time it takes to annotate just one cell image, for this study we limited our test set creation to two scenarios. We trained on datasets which contain numerous cell types, microscope modes, and magnifications. For this research we sought to prove efficacy for our segmentation algorithm using very difficult images to segment: brightfield images from a low cost microscope. We have run other types of images through SAMCell, and anecdotally, the segmentations of, for example, fluorescent images appear much more accurate than brightfield images.

4. A detailed discussion of the post-processing steps, including the rationale behind the watershed algorithm choice and potential alternatives, would strengthen the methodology section.

Thank you for this thoughtful suggestion. We have expanded our discussion of the watershed algorithm in Section 2.5.5, detailing its rationale and implementation for converting distance maps to cell masks. We've also added information about potential alternatives to our approach, and the rationale behind why they weren’t chosen.

Reviewer #2: The paper modifies Segment Anything Model (SAM) for cell segmentation task of phase-contrast and brightfield microscopic images. In addition they publish their own dataset two cell lines. I like the idea of using distance maps as prediction task, and evaluation on the own datasets.

Here are the comments:

1. I do not agree with the claim saying, "...being based on U-Net, the model may under-fit larger datasets and has a reduced advantage from pretraining.". U-Net is an architecture where any large backbone network can be used. If underfitting is the problem, the straightforward solution would be to simply increase the network size, use a larger CNN, add more layers, etc.

Thank you for the correction, we have removed this sentence from section 1. Because U-Net is a small CNN, we thought it was important to point out the potential of underfitting, but it is now made clear in the text that the issue is not inherent to U-Net, but rather a general consequence of CNN size.

2. In my opinion, the number of parameters is never one of the main advantages. "...SAMCell inherits Segment Anything’s main advantages: a high parameter, ..."

Thank you for sharing this thought. Given that it is subjective whether number of parameters is an advantage, we have rephrased this sentence in section 1 from “"...SAMCell inherits Segment Anything’s main advantages: a high parameter, Vision Transformer based architecture” to “..SAMCell inherits Segment Anything’s main advantage: a Vision Transformer based architecture” instead to focus solely on the ViT based architecture.

3. In the Introduction section, contributions of the paper should be highlighted.

1. Prediction of distance-transform map

2.Creation and publication of novel dataset.

This is an excellent note, we have made the contributions of this paper clear in the introduction by adding the following sentences “Our main contributions are twofold: First, we introduce a novel approach that fine-tunes SAM to output a real-valued distance map describing the Euclidean distance to a cell border for each pixel in the image. We then recover the boundary using a post-processing technique based on the watershed algorithm to effectively address the challenging problem of segmenting densely packed cells with ambiguous boundaries. Second, we create and publicly release two new annotated datasets, PBL-HEK and PBL-N2a, providing much-needed benchmark resources for evaluating cross-dataset generalization performance of cell segmentation algorithms across diverse cell morphologies and imaging conditions. These datasets contain phase-contrast microscopy images of cell lines commonly used in biological research but captured with different microscopes than those in existing training datasets, facilitating a realistic assessment of model generalization.” in section 1.

4. LIVECell (5000 images) and Cytoplasm (600 images) are not that large datasets.

Thanks for this comment. We have removed several instances where we refer to these datasets as ‘large’ and we have added the following clarification to the paper “While 5000 images may not be considered large in the context of…”. In summary, while LIVECell and Cellpose Cytoplasm are not large datasets in the context of a standard computer vision training dataset, they are unprecedently large data sets for the field of annotated cell images. At the time of model training, these are the only relevant datasets available with regards to developing a generalist model. Given that within each image there are hundreds of segmented cells, even though LiveCell only has 5000 images, there are 1.7 individual cells which the model is being trained on. For reference, it can take up to 20 minutes to annotate a cell image by hand. The lack of training data has been a significant issue in this field. This is one of our primary reasons for using SAM, which is already pre-trained, and does not require as much data for fine tuning tasks.

5. Figures in general lack details. The network details should be also visible from the architecture figure.

Thank you for this suggestion. The network architecture is not novel and was published in the original Segment Anything Model publication, which we have cited in the paper. We have also included a reference figure of SAM from the original paper. The intention is that we can abstract away the internal details of the Segment Anything Model, emphasizing instead the finetuning and the overall segmentation algorithm. To this end, we have added a detailed diagram of the SAMCell pipeline, complete with a view of our training data preparation, finetuning method, and inference using the watershed algorithm. We have added the following sentence in section 2.5 as well: “We retain the original architecture of the Segment Anything Model (SAM) and abstract away its internal details for simplicity.”

6. In the Sliding Window Approach, what is the reasons of cropping out 256x256px patches from the input images, then upsample to 1024x1024px? Is it better than directly upsampling the input images to 1024x1024px ?

Thanks for this question, in our new ablation study on patch sizes in appendix A.3, we have clarified that our approach maintains SAM's original input/output dimensions (1024×1024 input, 256×256 output) to leverage the pretrained model architecture without modifications. We have also added a deeper discussion on this topic in section 2.5.2 “We chose this sliding window approach over directly upsampling…”. For this work, we chose not to change the model architecture, and instead implemented a sliding window to restitch the images after the model ran.

7. What is the reasons to stop on 40th epoch? What if you train more?

Thank you for the comment. Upon further reflection, we agreed with the need for more precise understanding of training and model convergence. During the re-trainings conducted during this revision, we adopted a different training protocol as outlined in section 2.7 (Training Protocol). “We trained using early stopping over a range of 35 to 100 epochs, with a patience of 7 epochs and a minimum improvement threshold of 0.0001. We save the model after every checkpoint and take the version with the lowest loss…”. We found that for most configurations, our models stopped training after 30-35 epochs because of this criterion, proving that the model training had converged and training more would have no significant advantage. This resulted in higher perfomance than previously seen, so we would like to humbly thank our reviewer for driving this change.

8. The paper need more ablation studies:

1. Different sized patches.

2. Different threshold values in the post-processing.

3. Segmenting examples of different shaped cells (dividing, c-shape etc)

4. How the segmentations changed with each (or every 10) epochs?

Thank you for this suggestion. Upon further reflection, we agreed with the need for more precise ablation studies and actually upon performing them, we were able to optimize our parameters even further. This resulted in higher perfomance than previously seen, so we would like to humbly thank our reviewer for driving this change.

To this end, we have conducted a deeper pretraining ablation study, as well as four other ablation studies, all in appendix A. While we had done a course check for these parameters during our initial model training, we did a finer sweep and found some interesting results that we have added to the paper. Some of these changes have positively impacted the accuracy of our models as well, which are reflected in the updated metrics values. We have added an ablation study about different patch sizes, model variants, different threshold values in post processing, as well as different datasets for training. We have added how the segmentation metrics changed every 5 epochs in the pretraining ablation study for both random initialization and pretrained weights, giving interesting results. (appendix A.4). We have also added examples from PBL HEK and PBL N2a, which contain cell morphologies found in real scenarios, segmented using cellpose cyto (our best performing baseline) and SAMCell to understand how they differ qualitatively (section 3.5).

9. Table 1 seems unfair since SAMCell should be finetuned, and the baselines are trained from the scratch (or were they finetuned also?).

Thank you for the note. We have added this sentence in the caption "The Stardist, Cellpose, and CALT-US models were trained only on LIVECell-train then tested on LIVECell-test (top) and separately trained only on Cyto-train, then tested on Cyto-test (bottom). SAMCell, inheriting SAM's pretraining, was fine-tuned only on LIVECell-train then tested on LIVECell-test (top) and separately fine-tuned only on Cyto-train and then tested on Cyto-test (bottom)." To clarify, for table 1, the three baselines were trained on livecell and cyto from scratch and samcell was fine tuned using the same datasets. We have hypothesized that the reason samcell is able to get high accuracy is because of its architecture, and also its pretraining. This is one of the core advantages of using SAM - since it already has a prior for object segmentation, we should be able to get higher accuracy.

10. For Table 2, it is still better to show the scores of the baselines trained on the LIVECELL dataset. I believe they all have pretrained weights, also.

Thank you for this suggestion, unfortunately this is beyond the scope of this study. While this is a good idea, we ran into errors trying to load some of these weights. Addressing this would require significant training time and computational resources. We will consider pursuing these baselines for future work.

11. It is more comprehensive if segmentation comparisons from baselines and two-SAMCell models are shown.

Thank you for this note. For table 1, the values at the top represent all four models trained on LIVECell-train and then tested on the LIVECell-test datasets, and the values at the bottom represent all four models trained on Cyto-train and then tested on Cyto-test. This is to understand how well these models perform on data similar to the data they have encountered during training. This clarification has been added to the table caption, For Table 2, we now compare 3 variants of samcell with 3 of our baselines on zero shot datasets to understand each model’s performance with images it has never seen before.

12. User interface seems nice. It is also better to consider creating a plugin for image viewers, such as Napari or Imagej.

Thank you for this suggestion. We agree that a napari plugin would enhance the user experience even further since they will be able to manually correct any errors. To this end, we are developing a fully functioning plugin for napari called samcell-napari which is currently available on the napari hub, and is still being optimized. However, this is outside the scope of this study so we elect to omit this from the paper.

"Overall, the paper is well-written. However, there are still some gaps that need to be addressed to reproduce the results, especially inside the figures. In my opinion, there are not enough ablation studies to support the hyperparameter settings and other decisions, and the comparisons with the baselines are insufficient to support the claims. In its current version, this paper is more suitable to be submitted to a conference (could be top-tier).

We appreciate the suggestions and advice. We have made substantial revisions in response to the thoughtful suggestions of the reviewers, in order to meet your approval for publication in this journal. Once again, we humbly thank the reviewers for their insightful feedback as we work to publish this article.

---

## [Decision Letter · Decision Letter 1]

23 Jun 2025

PONE-D-25-05890R1SAMCell: Generalized Label-Free Biological Cell Segmentation with Segment AnythingPLOS ONE

Dear Dr. VandeLoo,

Thank you for submitting your manuscript to PLOS ONE. After careful consideration, we feel that it has merit but does not fully meet PLOS ONE’s publication criteria as it currently stands. Therefore, we invite you to submit a revised version of the manuscript that addresses the points raised during the review process.

We look forward to receiving your revised manuscript.

Kind regards,

Krishnendu Sinha, Ph.D.

Academic Editor

PLOS ONE

Journal Requirements:

Reviewers' comments:

Reviewer's Responses to Questions

**Comments to the Author**

1. If the authors have adequately addressed your comments raised in a previous round of review and you feel that this manuscript is now acceptable for publication, you may indicate that here to bypass the “Comments to the Author” section, enter your conflict of interest statement in the “Confidential to Editor” section, and submit your "Accept" recommendation.

Reviewer #2: All comments have been addressed

2. Is the manuscript technically sound, and do the data support the conclusions?

Reviewer #2: Partly

3. Has the statistical analysis been performed appropriately and rigorously? 

Reviewer #2: N/A

4. Have the authors made all data underlying the findings in their manuscript fully available?

Reviewer #2: Yes

5. Is the manuscript presented in an intelligible fashion and written in standard English?

Reviewer #2: Yes

6. Review Comments to the Author

Reviewer #2: Thank you the authors for addressing our comments.

1. One-sentence paragraphs are present; consider merging or expanding these for better readability and flow.

2. There are few figures with the same label.

3. Please specify the imaging modality for each dataset used in the paper — for instance, whether the data comes from fluorescence microscopy, IHC staining, or another technique. Possibly, report results on other modalities than phase contrast and light microscopy, or specifically mention that the methodology is suitable for those types of microscopic images.

4. In the DET evaluation, the term “empty graph” should be clarified.

5. Additionally, define AOGM-D explicitly so that readers unfamiliar with this metric can follow the evaluation logic.

6. The authors claim the MedSAM and SAMed models do not work well. But, they did not provide any evidence of this claim.

7. The authors mentioned problems in distance transform of peanut and C-shaped cells. They did not address the problem further. Are they removed from the training set, or are they the ones lowering the SEG and DET scores?

8. In the second paragraph of Section 2.5.2., why is there a need to upsample already large image?

9. In Section 2.5.5, it is confusing whether the watershed algorithm starts from the center of the cell or from the background. Authors mention that watershed starts from local minima.

10. The authors refer to MedSAM and SAMed, but does show the results. Results from those models should be included in the evaluation, or at least show an evidence that they do not segment correctly.

7. PLOS authors have the option to publish the peer review history of their article (what does this mean?). If published, this will include your full peer review and any attached files.

Reviewer #2: No

---

## [Author Response · Author response to Decision Letter 2]

18 Jul 2025

We thank the reviewer for their remaining feedback and diligence in reading our manuscript. We have read and carefully addressed each point raised, both below and in the text.

Thank you also to everyone involved in this process, including the other reviewers who did not have any requested revisions at this stage and the editor. We look forward to completing the peer review process and taking this manuscript across the finish line!

1. One-sentence paragraphs are present; consider merging or expanding these for better readability and flow.

Thanks for this note. We have merged any unintended one sentence paragraphs.

2. There are few figures with the same label.

Thanks for this comment. We are not sure which figures this refers to, as all of the figures have the correct label in our copy of the manuscript. It is possible that the compiling of the paper has caused you to see something different than we do. We will check with the editor to ensure that there is no mislabeling when the paper is published.

3. Please specify the imaging modality for each dataset used in the paper — for instance, whether the data comes from fluorescence microscopy, IHC staining, or another technique. Possibly, report results on other modalities than phase contrast and light microscopy, or specifically mention that the methodology is suitable for those types of microscopic images.

We appreciate the reminder. In this paper, our methodology is specifically optimized for phase-contrast and bright-field microscopy images, which represent the majority of label-free cell imaging applications and is the focus of our paper as a whole. We absolutely would consider publishing a paper that specifically considers fluorescently labeled datasets in the future.

We have ensured that the imaging modality is included for all the datasets mentioned in the paper. In summary, in the paper the following imaging modalities are specified:

LIVECell: "phase-contrast images"

Cytoplasm: "various microscopy techniques, including both bright-field and fluorescent images"

PBL-HEK and PBL-N2a: "phase-contrast microscopy images"

4. In the DET evaluation, the term “empty graph” should be clarified.

Thanks for this request for clarification. We now state that "empty graph" represents a graph with no nodes (no detected cells) by adding the following to Section 2.2:

“...an empty graph (a graph with no nodes, representing no detected cells) into…”

5. Additionally, define AOGM-D explicitly so that readers unfamiliar with this metric can follow the evaluation logic.

Thank you for the suggestion. We added an explicit definition of AOGM-D as quantifying the minimum cost of transforming one graph representation into another, by rephrasing the paragraph in Section 2.2 as follows:

“This is done with a graph-based approach, the “Acyclic Oriented Graph Matching Measure for Detection" (AOGM-D), which quantifies the minimum cost of transforming one graph representation into another through elementary graph operations. To compute the AOGM-D, we first generate graphs from the ground truth and model prediction, with each cell serving as a node. Then, the number of basic graph operations (e.g., add node, remove node, split node into two) required to convert the predicted graph into the ground truth graph is computed.}”

6. The authors claim the MedSAM and SAMed models do not work well. But, they did not provide any evidence of this claim.

Thanks for this comment, we will try to clarify. MedSAM and SAMed were not designed for cell segmentation—MedSAM targets medical imaging modalities like CT scans and X-rays, while SAMed focuses on organ segmentation in CT images. Therefore, directly applying these methods to cell segmentation would not be appropriate, and rationally would not be expected to perform even to the level of Default SAM(results shown in Figure 5). We refer to these works solely to present context for different fine-tuning strategies and architectural choices for adapting SAM to specialized domains.

More specifically,

MedSAM-style approaches would be impractical due to requiring bounding box prompts for hundreds of cells per image

SAMed-style categorical approaches cannot distinguish individual cell instances when cells are touching—a critical requirement for cell segmentation

We've enhanced the manuscript to clearly distinguish between the original applications of these methods and the theoretical limitations their strategies would face if adapted to cell segmentation. Our preliminary experiments with categorical segmentation approaches (inspired by SAMed's strategy) confirmed these theoretical limitations. We acknowledge that a comprehensive study of potential modifications to make these approaches suitable for cell segmentation could be valuable future work, but is beyond the scope of this study.

Added/Modified in Section 1:

“It is important to note that neither MedSAM nor SAMed were designed for cell segmentation, and therefore directly applying these methods to cell segmentation would not be appropriate or meaningful. However, both works give important insight into successful fine-tuning strategies and architectural choices for adapting SAM in a specialized domain. Inspired by MedSAM's encouraging segmentation performance of organs, we take a similar approach of fine-tuning the SAM image encoder for our particular dataset. However, we expand upon their work by eliminating the need for a bounding box prompt, which would be impractical for cell segmentation where hundreds of cells may be present in a single image. Similarly, while SAMed demonstrates semantic segmentation without prompts, we extend this approach by conducting unprompted instance segmentation rather than being limited to discrete categories, which cannot distinguish individual cell instances when cells are touching or overlapping.”

7. The authors mentioned problems in distance transform of peanut and C-shaped cells. They did not address the problem further. Are they removed from the training set, or are they the ones lowering the SEG and DET scores?

Thanks for your question. We expanded the discussion to clarify these are edge cases that may contribute to lower scores and were not removed from our realistic datasets.

Updated text in Section 2.4:

“This approach does have some drawbacks for cells consisting of multiple compartments. Namely, when a cell is peanut-shaped, like those formed when a cell is in the process of splitting, SamCell would consistently segment this shape as two cells, while an expert annotator may segment it either as one cell or two cells, depending on the person. While peanut-shape morphologies are a minority under typical cell culture conditions, we suspect that they may thus contribute to lower SEG and DET scores. For other non-circular shapes (like ovoid or "C-shaped" cells) we observe through empirical testing (data not shown) that the postprocessing technique can properly recover their geometries. It is important to note that our training and testing datasets contain realistic cell morphologies and expert annotations, without removing any problematic cases, ensuring our evaluation reflects real-world performance, so edge-cases like peanut-shaped cells are not removed from the dataset.”

8. In the second paragraph of Section 2.5.2., why is there a need to upsample already large image?

Good question! Thanks for highlighting this so we have the chance to address it. The primary reason is that SAM's architecture takes 1024×1024 inputs and produces 256×256 outputs. By using 256×256 patches, we can directly compare the original patch with the model's output, maintaining pixel-level correspondence for training and evaluation. Additionally, microscopy images are often much larger than SAM's required 1024×1024 input (e.g., 2048×2048 or larger). Direct resizing would cause distortions and loss of cellular detail through downsampling for large images. Our approach preserves original resolution within each 256×256 patch by consistently upsampling discrete regions to 1024×1024, maintaining the cellular features essential for accurate segmentation. We have clarified this in the text by adding/modifying the following in Section 2.5.2:

"We chose this sliding window approach over directly resizing the entire input image to 1024 x 1024 for two reasons. First, SAM's architecture takes 1024 x 1024 inputs and produces 256 x 256 outputs. By using 256 x 256 patches, we can directly compare the original patch with the model's output without further manipulation, maintaining pixel-level correspondence for training and evaluation. Second, microscopy images often contain thousands of pixels in each dimension (e.g., 2048 x 2048 or larger). Direct resizing of such large images to SAM's required 1024 x 1024 input could result in a catastrophic distortion or loss of cellular detail. Our sliding window approach preserves the original resolution by consistently upsampling 256 x 256 regions to 1024 x 1024, an approach which maintains the cellular features essential for accurate segmentation."

9. In Section 2.5.5, it is confusing whether the watershed algorithm starts from the center of the cell or from the background. Authors mention that watershed starts from local minima.

Thanks for this question. The algorithm treats our distance map as a topographical surface where cell centers are peaks and boundaries are valleys, starting from seed points(in the center of the cell) and then determining boundaries where flood regions from different seeds meet. We have clarified the watershed algorithm explanation to better describe how it operates in our specific context.

Added/Modified in Section 2.5.5:

“We apply the Watershed algorithm, a classical approach to boundary detection, to recover cell masks from the distance map. The algorithm treats the distance map as a topographical surface where cell centers represent peaks and cell boundaries represent valleys. Starting from the cell centers identified by the cell peak threshold, the algorithm simulates water flooding toward the simulated valleys. The cell boundaries are thus determined where the flood regions hit a clear boundary (the cell edge), or when the flood regions from different cell centers meet (if a cell edge is not clear), effectively separating individual cells even when they are densely packed….”

10. The authors refer to MedSAM and SAMed, but does show the results. Results from those models should be included in the evaluation, or at least show evidence that they do not segment correctly.

Thank you for this question, as your point is similar to that of Question 6, (The authors claim the MedSAM and SAMed models do not work well. But, they did not provide any evidence of this claim.) for brevity, we have addressed this point in our response to your question 6.

---

## [Decision Letter · Decision Letter 2]

13 Aug 2025

SAMCell: Generalized Label-Free Biological Cell Segmentation with Segment Anything

PONE-D-25-05890R2

Dear Dr. VandeLoo,

We’re pleased to inform you that your manuscript has been judged scientifically suitable for publication and will be formally accepted for publication once it meets all outstanding technical requirements.

Kind regards,

David Mayerich

Academic Editor

PLOS ONE

Additional Editor Comments (optional):

Reviewers' comments:

Reviewer's Responses to Questions

**Comments to the Author**

1. If the authors have adequately addressed your comments raised in a previous round of review and you feel that this manuscript is now acceptable for publication, you may indicate that here to bypass the “Comments to the Author” section, enter your conflict of interest statement in the “Confidential to Editor” section, and submit your "Accept" recommendation.

Reviewer #3: All comments have been addressed

2. Is the manuscript technically sound, and do the data support the conclusions?

Reviewer #3: Yes

3. Has the statistical analysis been performed appropriately and rigorously? 

Reviewer #3: Yes

4. Have the authors made all data underlying the findings in their manuscript fully available?

Reviewer #3: Yes

5. Is the manuscript presented in an intelligible fashion and written in standard English?

Reviewer #3: Yes

6. Review Comments to the Author

Reviewer #3: The authors have been responsive to previous reviewer feedback, resulting in an improved manuscript. SAMCell demonstrates solid technical merit for biological cell segmentation and addresses practical needs. The experimental validation is comprehensive, with cross-dataset evaluation showing robustness across different cell types and imaging conditions. The systematic comparison against established baselines, creation of evaluation datasets, user-friendly GUI, and open-source release will benefit the research community.

The main limitations include computational requirements (5 seconds per image on high-end GPU) that may restrict practical adoption, and relatively small evaluation datasets (5 images each) that somewhat limit the robustness of generalization claims. These limitations are acknowledged. This work is technically sound and practically useful.

7. PLOS authors have the option to publish the peer review history of their article (what does this mean?). If published, this will include your full peer review and any attached files.

Reviewer #3: No

---

## [Editor Report · Acceptance letter]

PONE-D-25-05890R2

PLOS ONE

Dear Dr. VandeLoo,

I'm pleased to inform you that your manuscript has been deemed suitable for publication in PLOS ONE. Congratulations! Your manuscript is now being handed over to our production team.

Kind regards,

on behalf of

Dr. David Mayerich

Academic Editor

PLOS ONE